# Comparative Interactome Analysis of Emerin, MAN1 and LEM2 Reveals a Unique Role for LEM2 in Nucleotide Excision Repair

**DOI:** 10.3390/cells9020463

**Published:** 2020-02-18

**Authors:** Bernhard Moser, José Basílio, Josef Gotzmann, Andreas Brachner, Roland Foisner

**Affiliations:** 1Max Perutz Labs, Center of Medical Biochemistry, Medical University of Vienna, Vienna Biocenter (VBC), 1030 Vienna, Austria; bernhard.a.moser@meduniwien.ac.at (B.M.); josef.gotzmann@meduniwien.ac.at (J.G.); 2Department of Vascular Biology and Thrombosis Research, Center for Physiology and Pharmacology, Medical University of Vienna, 1090 Vienna, Austria

**Keywords:** LEM-proteins, inner nuclear membrane, nuclear envelope, BioID, DNA repair, nucleotide excision repair

## Abstract

LAP2-Emerin-MAN1 (LEM) domain-containing proteins represent an abundant group of inner nuclear membrane proteins involved in diverse nuclear functions, but their functional redundancies remain unclear. Here, using the biotinylation-dependent proximity approach, we report proteome-wide comparative interactome analysis of the two structurally related LEM proteins MAN1 (*LEMD3*) and LEM2 (*LEMD2*), and the more distantly related emerin (*EMD*). While over 60% of the relatively small group of MAN1 and emerin interactors were also found in the LEM2 interactome, the latter included a large number of candidates (>85%) unique for LEM2. The interacting partners unique for emerin support and provide further insight into the previously reported role of emerin in centrosome positioning, and the MAN1-specific interactors suggest a role of MAN1 in ribonucleoprotein complex assembly. Interestingly, the LEM2-specific interactome contained several proteins of the nucleotide excision repair pathway. Accordingly, LEM2-depleted cells, but not MAN1- and emerin-depleted cells, showed impaired proliferation following ultraviolet-C (UV-C) irradiation and prolonged accumulation of γH2AX, similar to cells deficient in the nucleotide excision repair protein DNA damage-binding protein 1 (DDB1). These findings indicate impaired DNA damage repair in LEM2-depleted cells. Overall, this interactome study identifies new potential interaction partners of emerin, MAN1 and particularly LEM2, and describes a novel potential involvement of LEM2 in nucleotide excision repair at the nuclear periphery.

## 1. Introduction

In metazoan cells, the periphery of the nucleus is confined by the inner nuclear membrane (INM) and the underlying lamina, a filamentous meshwork of A- and B-type lamins [1]. The INM contains a diverse set of ubiquitous and tissue-specific integral membrane proteins, termed nuclear envelope transmembrane proteins (NETs) [2,3,4,5], but their functions are mostly elusive. The better characterized abundant NETs include a protein family defined by the presence of a LAP2-Emerin-MAN1 (LEM)-domain [6], which comprises seven members in mammals [7,8]. The LEM proteins emerin, several isoforms of lamina-associated polypeptide 2 (LAP2), MAN1, LEMD1 and LEM2 are integral membrane proteins of the INM [9] and interact with lamins in the nuclear lamina and with chromatin via binding of their LEM domain to the ubiquitous chromatin protein Barrier-to-Autointegration Factor (BAF) [10].

LEM2 (*LEMD2*) and MAN1 (*LEMD3*) are closely related LEM proteins containing an N-terminal LEM domain, two transmembrane domains and a MAN1-SRC1p-C terminal (MSC) domain (Figure 1A) [11]. MAN1 contains an additional RNA recognition motif (RRM) at the C-terminus. Emerin (*EMD*) is the smallest abundant LEM protein with an N-terminal LEM domain and one C-terminal transmembrane domain (Figure 1A) [7]. These LEM proteins serve numerous unique and redundant functions in nuclear architecture and mechanics [7,8,9,12] and in chromatin organization [13,14,15]. In addition, they are involved in the regulation of various signaling pathways. Emerin affects the wnt, Notch and Rb/MyoD pathways [16,17,18], MAN1 inhibits transforming growth factor beta- (TGF-ß) and bone morphogenic protein (BMP) signaling by recruiting R-Smads to the nuclear periphery [19,20,21], and LEM2 regulates MAP and AKT kinases [22]. Interestingly, several studies on the proteins’ yeast orthologues and a few recent studies in metazoan cells indicate that LEM proteins may be involved in DNA damage repair, particularly the repair of double strand breaks [23,24,25,26,27,28], but the underlying mechanisms are still unclear. In *Caenorhabditis elegans baf-1*, *lem-2* (Ce-MAN1) and *emr-1* (Ce-emerin) mutants were found to be hypersensitive to DNA damage-causing ionizing radiation [29]. Mammalian BAF, a common binding partner of LEM proteins, interacts with the damage-specific DNA binding proteins (DDB) 1 and 2, and additionally, BAF and emerin associate with DDB2 and Cul4 after UV irradiation [30].

There are several studies in various metazoan model systems indicating that emerin, LEM2 and/or MAN1 have redundant functions in vivo. In *C. elegans*, single *emr-1* and *lem-2* mutants have a mild phenotype, while double mutants display severe developmental and muscle defects [31]. Also in *Drosophila*, only double LEM protein mutants have reduced viability [32], and mammalian emerin and LEM2 have overlapping functions in regulating extracellular signal-regulated kinase (ERK) signaling [22]. On the other hand, these LEM proteins were reported to have several unique functions, including the regulation of diverse signaling pathways [9]. Furthermore, numerous point mutations have been identified in human genes encoding emerin, LEM2 and MAN1, which cause highly divergent phenotypes, ranging from heart defects and muscular dystrophies to bone and cartilage pathologies, collectively called nuclear envelopathies [33]. Mutations in emerin lead to Emery–Dreifuss muscular dystrophy [34], MAN1 mutations cause disorders affecting osteogenesis via deregulated BMP signaling [35], and LEM2 has been linked to Hutterite-type cataracts [36], arrhythmic cardiomyopathy [37] and LEMD2-associated nuclear envelopathy with progeria-like phenotype [38]. Furthermore, while emerin-knockout mice show a mild muscle phenotype [18], *Lem2* deficiency is embryonic lethal in mice [39].

Although our knowledge about LEM proteins is steadily improving, their unique and redundant functions are still incompletely understood. In order to shed light on this open question we performed comparative interactome analysis of emerin, MAN1 and LEM2. Commonly, these proteins localize at the inner nuclear membrane and interact with lamin A/C and BAF [10,11,40,41]. Using a proximity-dependent biotin labelling method (BioID) [42], we identified previously reported as well as novel interaction partners of emerin and MAN1 and show, for the first time, a comprehensive LEM2-specific interactome. Surprisingly, several components of the nucleotide excision repair (NER) machinery were specifically identified in the LEM2 interactome, indicating a putative novel function of LEM2 in the NER pathway. Accordingly, LEM2, but not emerin or MAN1 depletion reduced cell viability following UV treatment.

## 2. Materials and Methods

### 2.1. Cell Culture and In Vivo Biotin Labeling

Human epithelial bone osteosarcoma cells U2OS were cultured in Dulbecco’s Modified Eagle Medium (DMEM) (GE Healthcare, Illinois, Chicago, USA) supplemented with 2 mM L-Glutamine, nonessential amino acids, 100 U/ml penicillin and streptomycin and 10% fetal calf serum (Invitrogen, Vienna, Austria) at 37 °C and 8.5% CO_2_. Cells stably expressing doxycycline-inducible LEM proteins fused to BirA* biotin ligase or BirA*-GFP as a control were selected with 100 µg/ml hygromycin. For protein labeling, U2OS cells were incubated for 6 h in complete medium containing 1 µg/mL doxycycline in 150 mm dishes before they were washed with 1 x DPBS and incubated for additional 16 h with or without 50 µM biotin. For gene knockdown by RNA interference, cells were seeded in complete DMEM 24 h prior to transfection with 5 µM siRNA pools (Dharmacon/GE, Lafayette, Colorado, USA) using 5µl DharmaFECT (Thermo Scientific, Vienna, Austria) according to the manufacturer’s instructions. Two transfections at an interval of 24 h were performed, and cell culture medium was changed each time. For analysis of survival and cell growth of UV-C-treated cells, 5 × 10^4^ cells per well were seeded into a 6-well multidish and allowed to attach overnight, before cells were washed once with 1 × DPBS, covered with 500 µl 1 × DBPS and exposed to 5 J/m^2^ with a Stratalinker 2400 (λ 254 nm) UV crosslinker (Stratagene, Santa Clara, California, USA) and subsequently cultured in complete DMEM for 96 h.

### 2.2. Plasmids and siRNAs

To create plasmids expressing BirA*-fusion proteins, BirA* cDNA [42] was amplified by PCR with primers containing an SfiI restriction site at the 5’ end and subsequently ligated into the pJET1 vector (Thermo Fisher Scientific, Vienna, Austria). A Gateway^®^ (GW, Invitrogen, Vienna, Austria) cassette including a V5- and 6x histidine tag was amplified by PCR using primers containing a SfiI restriction site at the 3´ end from pDEST (Invitrogen, Vienna, Austria) and ligated into pJET1-BirA*. The BirA*-GW sequence was excised from pJET-BirA*-GW using the restriction enzyme SfiI and ligated into the pEMI vector using the same restriction sites. Previously generated pEntry vectors containing cDNAs coding for LEM2, MAN1, emerin and GFP were used to generate pEMI-BirA*-LEM2, pEMI-BirA*-MAN1, pEMI-BirA*-emerin and pEMI-BirA*-GFP by GW cloning, according to the manufacturer’s instructions. All sequences were verified by sequencing prior to experimental use. For gene knockdown by RNA interference of *LEMD2, LEMD3, EMD* and *DDB1*, the following siRNAs were used (Dharmacon Inc., Lafayette, Colorado, USA): SMARTpool: ON-TARGETplus Human LEMD2 (L-017941-02-0005), SMARTpool: ON-TARGETplus Human LEMD3 (23592) (L-006306-01-0005), SMARTpool: ON-TARGETplus Human EMD (2010) (L-011025-00-0005), SMARTpool: ON-TARGETplus Human DDB1 (1642) (L-012890-00-0005), ON-TARGET plus Non-targeting Pool (D-001810-10-20).

### 2.3. BioID Pull-Down and Preparation for Mass Spectrometry

U2OS cell lines, stably transfected with inducible BirA*-LEM2, BirA*-MAN1, BirA*-emerin and BirA*-GFP, were incubated as described above. After three washing steps with 1 × DPBS, cells were scraped off and lysed on ice in (1 ml / 15 mm petri dish) high-salt RIPA buffer (25 mM Tris-HCl pH 7.4, 500 mM NaCl, 1% NP-40, 1% sodium deoxycholate, 0.1% SDS), supplemented with 1 x complete protease inhibitor mix (Roche, Vienna, Austria). After sonication (Bandelin Sonopuls HD200; power MS 73/D, continuous pulse), 100 µl Pierce streptavidin magnetic beads (Thermo Scientific, Vienna, Austria) were added to the lysate and incubated overnight at 4 °C. Magnetic beads were collected and washed extensively with the following washing buffers in sequence—twice with high-salt RIPA buffer, twice with wash buffer A (2% SDS in ddH_2_O), once with wash buffer B (0.1% deoxycholate, 1% Triton X-100, 500 mM NaCl, 1 mM EDTA, and 50 mM Hepes, pH 7.5), once with wash buffer C (250 mM LiCl, 0.5% NP-40, 0.5% deoxycholate, 1 mM EDTA, and 10 mM Tris, pH 8.1), twice with wash buffer D (50 mM Tris, pH 7.4, and 50 mM NaCl) at 4 °C. 10% of the samples were used for Western blot analysis and 90% were analyzed by mass spectrometry after on-bead tryptic digest.

### 2.4. Western Blotting

For protein analysis, cells were lysed in high-salt RIPA buffer supplemented with 1x complete protease inhibitor mix (Roche, Vienna, Austria). Protein concentrations were measured by Bradford assay and equal amounts (10 µg total protein) were separated by SDS-PAGE and transferred onto nitrocellulose membrane (GE Healthcare, Illinois, Chicago, USA). Afterwards, the nitrocellulose membranes were blocked in 5% low-fat milk powder (Roth, Karlsruhe, Germany) in PBS containing 0.025% Tween20, incubated for 1 h with primary antibodies and for 1 hour with secondary horseradish peroxidase (HRP)-coupled antibodies (Jackson ImmunoResearch, West Grove, Pennsylvania, USA). After the first and second antibody incubation, membranes were washed three times with PBS containing 0.025% Tween20. Finally, the membranes were incubated with Amersham ECL prime (GE Healthcare, Illinois, Chicago, USA) or West Pico chemiluminescent substrate to detect signals by exposure to light-sensitive films (Thermo Scientific, Vienna, Austria). Band intensities were quantified with ImageJ (NIH, Bethesda, Maryland, USA), normalized to Ponceau S values and presented as fold increase (UV treated/untreated). Primary antibodies used for Western blot analysis were LEM2 (017340, Human Protein Atlas, Stockholm, Sweden), emerin (8A1, a gift from Glenn E. Morris), MAN1 (GTX84220, GeneTex, Irvine, California, USA), γ-tubulin (MFCD00677366, Sigma-Aldrich, Vienna, Austria), DDB1 (GTX100232, GeneTex, Irvine, California, USA), γH2AX (05-636, Upstate, Vienna, Austria) and anti-V5 tag (R960-25, Invitrogen, Vienna, Austria). The HRP-coupled Streptavidin was purchased from Invitrogen (SN1004).

### 2.5. Immunofluorescence

U2OS cells were grown on glass coverslips and fixed in 4% paraformaldehyde/PBS for 10 minutes at room temperature, permeabilized for 5 minutes in PBS containing 0.5% Triton X-100 and blocked by incubation in IF blocking buffer (PBS containing 0.5% cold water fish gelatin (Sigma, Vienna, Austria)) for 15 minutes. All primary and secondary antibody dilutions were prepared in IF blocking buffer and applied for 60 minutes. DNA was stained with 1 µg/ml DAPI (Sigma, Vienna, Austria) and coverslips were mounted on microscopy slides in Mowiol (Sigma, Vienna, Austria). All samples were imaged using a confocal laser scanning microscope (LSM-Meta 510, Carl Zeiss AG, Oberkochen, Germany) using a Plan-Apochromat 63x/1.4 Oil DIC objective. Primary antibodies used for immunofluorescence were the same as used for Western Blotting. In addition, an Alexa Fluor™ 647-Streptavidin (016-600-084, Jackson ImmunoResearch, West Grove, Pennsylvania, USA) was used for detection of biotinylated proteins. Secondary antibodies conjugated to Alexa-488 were purchased from Molecular Probes and secondary antibodies conjugated to Cy5 or TexasRed from Jackson ImmunoResearch (West Grove, Pennsylvania, USA).

### 2.6. Mass Spectrometry

Sample processing, mass spectrometric analysis and raw data preparations were performed by the Mass Spectrometry Facility at the Max Perutz Labs Vienna. BioID samples were analyzed using a Q Exactive Hybrid Quadrupole Orbitrap Mass Spectrometer (Thermo Scientific, Vienna, Austria).

### 2.7. Mass Spectrometric Data Analysis

For BioID, a total of 11 control runs (-Doxycycline) conducted on U2OS cells expressing BirA*-emerin-V5, BirA*-LEM2-V5 and BirA*-MAN1-V5 were analyzed. Furthermore, four control samples and three BioID runs conducted on U2OS cells expressing BirA*-GFP-V5 were analyzed. Mass spectrometric data were analyzed with SAINT v. 2.5.0 (Le Petit-Quevilly, France) using as options GSL_RNG_SEED = 123, sampling iterations for the burn-in period of Markov chain Monte Carlo (MCMC) = 2000, number of sampling iterations in MCMC = 10000, lowMode = 1, minFold = 1 and normalize = 1. SAINT v2.5.0 (Le Petit-Quevilly, France) allows for the application of a normalization step that divides spectral counts by the total spectral counts in each purification. SAINT v. 2.5.0 (Le Petit-Quevilly, France) REFORMAT script was used to reduce the control data into the four highest control counts [43,44,45]. High-confidence interactors were selected based on an Average Probability (AvgP) of ≥0.45. All GFP interactors (AvgP ≥ 0.45) were removed from the interaction results obtained when analyzing the interactors of emerin, LEM2 and MAN1. Data were visualized using ProHits-Viz (Toronto, Canada) [46]. Interaction data were downloaded from the STRING database v. 11.0 [47] and modified using Cytoscape software v. 3.7.1 (Seattle, Washington, USA) [48]. Gene ontology (GO) enrichment analysis was performed using ENRICHR, considering a fold change ≥1 and *p* value ≤ 0.05 [47,49]. Sequence homology was analyzed using Protein Blast (NCBI).

### 2.8. Statistical Analyses

Error bars represent standard error of the mean of at least eight independent viability assay experiments. Statistical significance was calculated by two-way ANOVA and Tukey’s multiple comparison. *P*-values smaller than 5% were considered statistically significant and indicated with an asterisk (‘*’ for *p* < 0.05; ‘**’ for *p* < 0.01; ‘***’ for *p* < 0.001; ‘****’ for *p* < 0.0001).

## 3. Results

### 3.1. Identification of the Interactomes of Emerin, MAN1 and LEM2 by BioID

We aimed at defining the shared and specific interactomes of emerin, LEM2 and MAN1 (Figure 1A) using the proximity-dependent biotin identification (BioID) approach [42]. This unbiased method is based on the nonreversible in vivo tagging of potential interaction partners of the protein of interest by the promiscuous *Escherichia coli* biotin ligase, BirA*, and allows detection of weak and transient interactions. Furthermore, BioID is preferable over protein coprecipitation approaches for proteins embedded in rigid and insoluble cellular structures, such as the nuclear lamina.

We generated stable U2OS cell lines, allowing doxycycline-inducible expression of V5-tagged BirA*-LEM-fusion proteins: BirA*-emerin-V5, BirA*-LEM2-V5, BirA*-MAN1-V5 (Figure 1A,B) and BirA*-GFP-V5. In the absence of doxycycline, BirA*-tagged constructs were undetectable by immunofluorescence microscopy (Figure 1C) and by immunoblotting of total cell lysates (Figure 1D) using an anti-V5 antibody. Addition of 1 µg/ml doxycycline to the medium for 22 h yielded a robust expression of BirA*-LEM proteins, which localized correctly at the nuclear rim, and for emerin, additionally in the endoplasmic reticulum (Figure 1C). Furthermore, in the presence of 50 μM d-Biotin for 16 h, we observed a strong increase in the amount of biotinylated proteins both in immunofluorescence using fluorescently labeled streptavidin (Figure 1C), and in immunoblotting using streptavidin-HRP (Figure 1D). The cellular localization of biotinylated proteins reflected the cellular distribution of the BirA*-LEM fusion constructs, mostly at the nuclear periphery for MAN1 and LEM2 and for emerin, additionally in the endoplasmic reticulum. Minor signals in the nucleoplasm may be due to transient interactors, which shuttle between the nuclear envelope and the nucleoplasm. Overall, these data indicate biotinylation of potential LEM protein interaction partners preferentially at the inner nuclear membrane.

To determine the interactomes of emerin, MAN1 and LEM2, we performed four independent BioID experiments each and 11 uninduced (-Doxycycline) control samples in total, followed by mass spectrometric analyses of biotinylated proteins. In addition, we used four control samples and three BioID runs conducted on U2OS cells expressing BirA*-GFP-V5 and subtracted these data sets from those of the BirA*LEM protein fusions. In order to identify high-confidence interactors, we analyzed mass spectrometry hits using SAINT v. 2.5.0 [43,44,45]. From the SAINT output we generated a score that rates the potential interactors based on an average probability of ≥ 0.45, yielding a total of 629 high confidence binding partners out of a total of 9990 hits; 44 high confidence hits for emerin, 94 for MAN1 and 491 for LEM2 (Figure 2A, Appendix A). The low number of high-confidence emerin interactors (44) may be due to the low expression level of the BirA*-emerin fusion protein relative to the other baits (Figure 1D, Appendix A). However, MAN1 and LEM2 fusion proteins were expressed at similar levels (Appendix A), yet the LEM2 interactome was about five times larger than that of MAN1 (491 versus 94, Figure 2A), making it difficult to compare them. Nevertheless, 59% of emerin interactors and 66% of MAN1 interactors were also detectable in the LEM2 interactome (Figure 2A, Appendix A). The extent of similarity of interactomes was also shown by calculating the distance matrices using ProHits-Viz [46] (Figure 2B). Interestingly, while only around 30% of the emerin and MAN1 interactomes were unique for these proteins, 86% of the large LEM2 interactome was unique for LEM2, making it particularly interesting for identifying novel LEM2 interactors (Figure 2A).

In an attempt to get some insight into common and unique functions of these proteins, we performed gene ontology (GO) enrichment analysis (fold change ≥ 1, *p*-value ≤ 0.05) to reveal significantly enriched GO terms related to biological processes [47,48,49]. GO enrichment analysis confirmed the multiple functions of LEM proteins and revealed a diverse list of biological processes encompassing nuclear envelope and chromatin-linked processes, such as “protein complex assembly”, “nuclear envelope organization” and “cell cycle and mRNA regulation”, among several others in both the common and unique hits. Interestingly, GO term enrichment in the emerin-specific interactome was consistent with previously reported specific functions of this protein (Figure 2C). Among the most significant GO terms (adjusted p value: 1.10E-03) among emerin-associated proteins were “centrosomal and microtubule-based processes” and “G2/M transition of mitotic cell cycle” (Figure 2C, Table 1). This result conforms to a previous study demonstrating the presence of emerin also in the outer nuclear membrane, where it is involved in the microtubule-mediated attachment of centrosomes to the nucleus [50]. Emerin-interacting proteins within this GO term include serine/threonine-protein kinases (PLK1), implicated in centrosome maturation and spindle assembly, centrosomal proteins (CEP131, CEP135) and microtubule-associated proteins (KIF14) (Figure 2C,D and Table 1). Thus, our analyses revealed several potential new binding partners for emerin, which may reflect its unique role in linking the centrosome to the outer nuclear membrane. For MAN1 interaction partners, “Ribonucleoprotein complex assembly” was one of the most significant GO terms (adjusted *p*-value: 01.00 E-02) (Figure 2C,D and Table 1), including PRPF19, SART1, CDCL5, SF1 and SNW1. The latter is, like MAN1, implicated in TGF-β signaling. However, the physiological relevance of these interactions remains to be tested.

### 3.2. High Representation of Nucleotide Excision Repair Proteins in the LEM2 Interactome

Unlike for MAN1 and emerin, very little is still known about binding partners and functions of LEM2, making it particularly interesting to analyze the LEM2-specific interactome. Our bioinformatical analysis of LEM2 data unexpectedly, but consistently, revealed several components of the nucleotide excision repair (NER) pathway among the high confidence hits of LEM2 interaction partners (Figure 2C and Figure 3A, Table 1). Furthermore, GO term enrichment analysis revealed that nine out of 421 hits were associated with transcription-coupled nucleotide excision repair (adjusted *p*-value: 2.46E-05; 7.57 fold enrichment) (Figure 3A,B). Among the highest scored proteins of the LEM2 interactome was the DNA damage-binding protein 1 (DDB1), which represents the large subunit (p142) of the DNA damage recognition complex of the NER machinery. DDB1 forms a complex with DDB2 (p48) and binds to specific types of DNA lesions, followed by recruitment of additional proteins required for DNA repair (Figure 3B) [51]. Besides DDB1, we found several downstream components of the NER pathway in the top candidate interactor list for LEM2, including COPS4 and GPS1 of the COP9 signalosome complex, which regulates the activity of DDB1, the ubiquitin ligase Cullin4A (CUL4A) and some NER-linked transcriptional proteins like the DNA helicase ERCC6L, some polymerase subunits (POLD2, POLR2B, POLR2C, POLR2E), and the elongation factor RFC3 (Figure 3A,B and Table 1). These data provide strong evidence for LEM2 being a novel component involved in NER.

### 3.3. Reduced LEM2 Levels Increase Sensitivity of Cells to UV-induced DNA Damage

In order to corroborate a potential involvement of LEM2 in the NER pathway, we investigated the ability of LEM2-depleted cells to respond to DNA damage in vivo. LEM2, emerin or MAN1 levels were significantly reduced by siRNA-mediated knockdowns in parental U2OS cells. Knockdown efficiency was confirmed 72 h after transfection by immunofluorescence (Figure 4A) and immunoblot analysis (Figure 4B). In addition, we also knocked down DDB1, a key NER protein, as a positive control in our assays. To test the ability of cells to cope with DNA damage, we irradiated them with a sublethal dose of UV-C (254 nm) and compared their proliferation over a time period of 96 h. UV-C treatment preferentially induces purine nucleotide dimerization, for the repair of which a functional NER is particularly important [51,52].

In the absence of UV-C radiation (negative control), DDB1-depleted cells exhibited a reduced cell proliferation compared to emerin-, LEM2-, MAN1- and mock-depleted (validated nontargeting RNAi) cells, owing to the central function of DDB1 in NER (Figure 4C,D). Following irradiation of cells with UV-C 48 h after transfection with the respective siRNAs, DDB1-depleted cells showed the strongest proliferation defect compared to the scrambled RNAi control. In contrast, emerin and MAN1-depleted cells proliferated like control cells, while LEM2-depleted cells showed an impaired cell growth recovery and a significant reduced cell count after UV damage (Figure 4C,D). In order to test the response of cells to DNA damage biochemically, we analyzed the increase of phosphorylated H2A.X (γH2A.X) protein levels, a common marker for elicited DNA damage response, by immunoblotting 48 h after UV-C treatment (Figure 4E,F). While control cells (scrambled RNAi) did not show an increase in γH2AX levels, indicating that those cells efficiently repaired DNA damage within 48 h, emerin and MAN1-depleted cells showed only a subtle increase (<1.5-fold, Figure 4E,F). In contrast, LEM2-depleted (2.5-fold) and DDB1-depleted (5-fold) cells showed substantially higher γH2AX levels after 48 hours (Figure 4E,F), indicating that their DNA damage repair was delayed or impaired. Overall, our data point towards a novel specific role of LEM2 in NER.

## 4. Discussion

In this study we report a proteome-wide comparative interactome analysis of three related LEM proteins using the BioID proximity labeling approach. BioID allows for the identification of stable and transient, and direct and indirect proximal (within 10 nm) protein-protein interactions under physiological conditions [42]. We demonstrate that all three BirA*-LEM proteins are nontoxic and localize correctly at the inner nuclear membrane following expression in the cell line U2OS. U2OS cells have frequently been used to investigate DNA repair pathways including NER (see for instance [53,54,55,56,57,58,59]). Mass spectrometric analyses of biotinylated proteins revealed several novel, as well as previously reported, binding partners of emerin, LEM2 and MAN1.

Detailed comparative analyses of the interactomes of the three LEM proteins turned out to be difficult due to the different numbers of high-confidence hits in these interactomes. The low expression of BirA*-emerin fusion protein compared to those of LEM2 and MAN1 fusions is likely the reason for the relatively low number (44) of high-confidence hits. MAN1 and LEM2 fusion proteins, however, were expressed at similar levels, yet the LEM2 interactome included ~five times more hits than that of MAN1 (491 versus 94 candidates). As a possible explanation, we can only speculate that the LEM2 fusion protein may be more accessible than the MAN1 protein, giving rise to an increased number of (probably) dynamic interactions. Based on the fact that around 60% each of the emerin and MAN1 interactors are also detected in LEM2, one may conclude that all three proteins are engaged in similar complexes, but LEM2 in addition has a large number of unique (probably dynamic) interactions. One concern in proximity-based labeling is the identification of unspecific hits, which we cannot completely exclude. However, using two different controls we tried to minimize the risk of these unspecific hits. One control (-Doxycycline) is expected to remove biotinylated endogenous proteins, while the BirA*-GFP negative control should eliminate so-called “sticky” proteins, which are frequently found as contaminants in mass spectrometry studies [60]. The identification of several previously reported binding partners of the LEM proteins support the validity and specificity of our interactome analysis.

While the raw data set included several known common binding partners of these proteins at the NE, such as lamins, inner nuclear membrane proteins and nuclear pore proteins, many were not among the high-confidence hits following the very stringent bioinformatic analyses. However, we identified several previously reported binding partners unique for emerin in our analysis. Among prominent hits for emerin interactors were several centrosomal proteins. A previous report showed that centrosomes were located further away from the outer nuclear membrane in emerin-depleted versus control cells (3.0 μm versus 1.5 μm) and emerin was demonstrated to interact with β-tubulin-containing centrosomes [50], but potential candidate proteins involved in the emerin-centrosome connection remain unknown. Our BioID-data identified several centrosomal- or microtubule-associated proteins implicated in diverse centrosome-linked functions like centriole biogenesis and localization, microtubule stability and polymerization, and mitotic spindle formation. BioID-based identification of emerin binding partners does not prove direct interactions with these proteins, though it demonstrates that they are in close proximity to emerin. In the MAN1-specific interactome, we identified a number of proteins implicated in ribonucleoprotein complex assembly, which supports a possible role of MAN1 in pre-mRNA processing. In line with these data, MAN1 was identified as a candidate heterodimeric splicing factor U2 snRNP auxiliary factor (U2AF) homology motif-containing protein, which plays important roles during RNA splicing [61]. However, the physiological relevance of these MAN1 interactions remains to be determined.

Compared to emerin and MAN1, very little is known about LEM2-specific interactors. For this reason, and because the LEM2 interactome was the largest in our analyses, we decided to particularly focus on the functional analyses of the LEM2-specific hits. Surprisingly, the LEM2-specific high-confidence binding partners revealed in this study include several components of the NER pathway, indicating a novel role of LEM2 in DNA damage repair. In support of this novel function of LEM2, we demonstrate that unlike MAN1- and emerin-depleted cells, LEM2-depleted cells showed significant impairment of proliferation following UV-C-mediated DNA damage. Interestingly, several recent studies reported a potential link between LEM proteins and DNA damage repair pathways [28,29,30,62,63,64], but molecular insight into these potential functions has not been reported yet. The NER machinery is a major repair mechanism in all organisms that includes transcription-coupled nucleotide excision repair (TC-NER) and global genome nucleotide excision repair (GG-NER) in nontranscribed or transcriptionally silent DNA [65]. NER handles a variety of DNA lesions including 6-4 pyrimidine-pyrimidone photoproducts and cyclobutane-pyrimidine dimers (UV irradiation), 8,5′‑cyclopurines-2′-deoxynucleosides (oxidative stress) or lesions which are induced by chemotherapeutic drugs [51,52]. The first hint that LEM2 may be involved in NER came from studies on the LEM2 orthologue in *Schizosaccharomyces pombe*. Yeast lem2 mutants presented the same hydroxyurea-induced cell cycle arrest phenotype as ddb1 mutants, an essential component of NER [66]. Our LEM2 interactome analysis identified various proteins of the GG-NER and TC-NER recognition complex. Together with the impaired response of LEM2-depleted cells to UV-C irradiation, these data provide strong evidence for the involvement of LEM2 in NER. To this end, we can only speculate about the specific LEM2-mediated functions in NER, but there is increasing evidence that, at least in some organisms, DNA repair occurs at the nuclear periphery [23,24,25,26,27,28] and several nuclear envelope proteins were implicated in DNA repair pathways at different levels. Lamin A/C was found to bind to and stabilize 53BP1 [67] and the inner nuclear membrane proteins SUN1 and SUN2 recruit the DNA-dependent protein kinase (DNAPK) complex [68]. LEM2 may similarly recruit, stabilize and/or orchestrate NER proteins at DNA lesions located at the nuclear periphery. Alternatively, LEM2 depletion may only indirectly affect NER through reorganization of chromatin and changes in gene expression of NER components as shown for lamin B1 depletion [62], or by affecting positional stability of repair foci as shown for lamin A deficiency [69].

In conclusion, our study revealed a comprehensive interactome of the three INM LEM proteins emerin, MAN1 and, particularly, LEM2, revealing common and unique interaction partners of the proteins, further supporting the role of emerin in centrosome positioning, and suggesting a potential involvement of MAN1 in ribonucleoprotein assembly. Most importantly, however, our analyses report for the first time a large interactome of LEM2 and identifies a possible novel role of LEM2 in NER, as also supported by our functional analysis. Thus, our data provide a valuable resource for performing further research on the regulation and functions of emerin, MAN1 and, particularly, LEM2.

## Figures and Tables

**Figure 1 cells-09-00463-f001:**
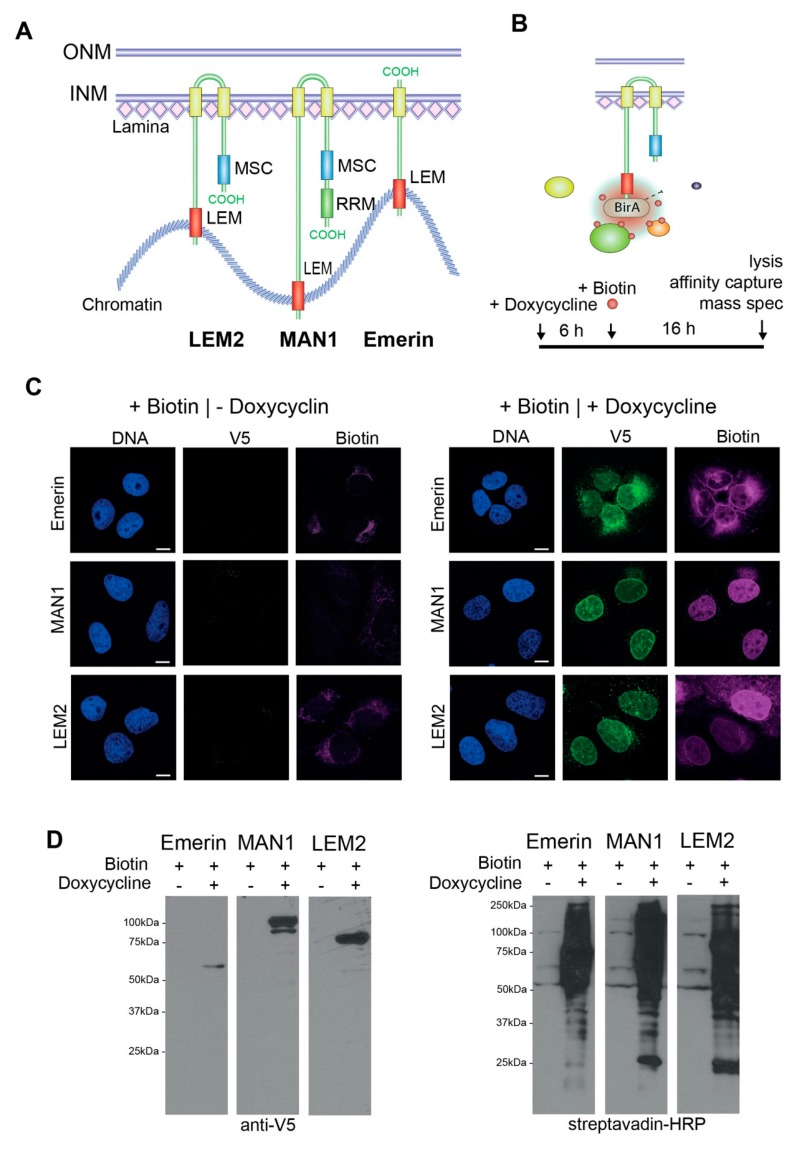
Expression of BirA*-emerin-V5, BirA*-MAN1-V5 and BirA*-LEM2-B-V5 in U2OS cells. (**A**) Schematic domain organization of LEM2-, MAN1- and emerin. Yellow, transmembrane domain; blue, MSC- (MAN1-SRC1p-C-terminal) domain; red, LEM-domain; green, RNA recognition motif (RRM). (**B**) BirA* fusion proteins biotinylate proteins in close proximity (10 nm). Cartoon and time course showing the BioID approach. Cells were incubated with medium containing 1 µg/ml doxycycline to induce fusion protein expression, 50 µM biotin was added after 6 h and incubated for another 16 h before cells were lysed. (**C**) U2OS cells stably expressing doxycycline-inducible BirA*-fusion constructs were incubated without (-Doxycycline) or with (+Doxycycline) doxycycline and biotin and processed for immunofluorescence microscopy using antibodies to the V5 tag and fluorescently labeled streptavidin (biotin). DNA was stained with DAPI. Bar, 10 µm. (**D**) Total cell lysates of incubated cells were prepared and analyzed by immunoblotting using V5-antibody and horseradish peroxidase (HRP)-conjugated streptavidin. Lanes in Figure 1D were on the same blot (see Appendix A for unedited immunoblot) and were generated with identical exposure time.

**Figure 2 cells-09-00463-f002:**
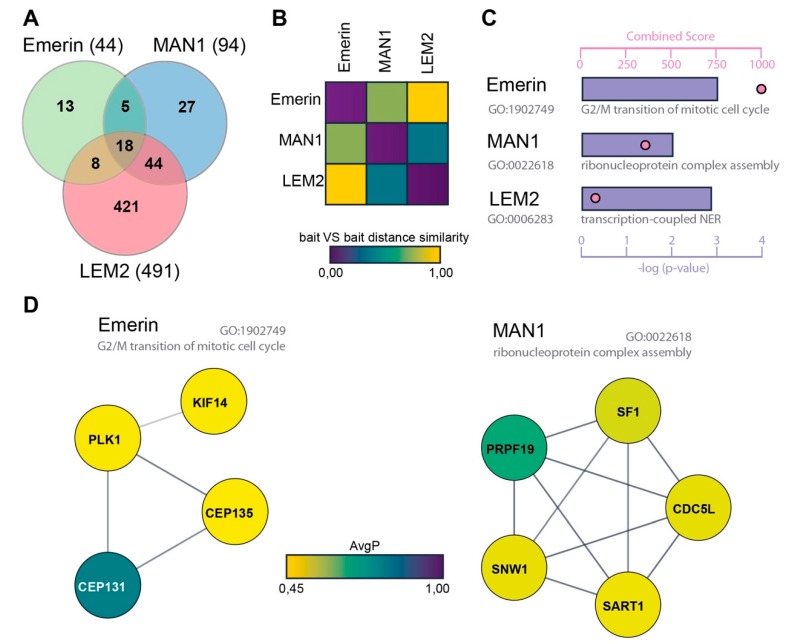
BioID identifies common and specific interactors for emerin, MAN1 and LAP2-Emerin-MAN1 (LEM2). Mass spectrometric data sets were analyzed using SAINT v. 2.5.0 [43,44] based on an average probability (AvgP) of ≥ 0.45. (**A**) Venn diagram showing the overlap of interactomes. Numbers indicate identified high-confidence interactors. (**B**) Bait vs. bait distance similarity analysis revealed a larger similarity between binding partners of MAN1 and LEM2 than the similarity of those proteins with emerin. (**C**) Gene ontology (GO) enrichment analysis using ENRICHR, considering a fold change ≥ 1 and *p*-value ≤ 0.05 [47]. ENRICHR identified enriched biological processes for each LEM protein. Blue bars represent *p*-value (log10) and pink points show GO term fold enrichment. (**D**) The identified interactors of emerin and MAN1 of indicated GO biological processes were further grouped and analyzed using the STRING database. The fill color of the nodes represents the AvgP and the line width represents the STRING interaction.

**Figure 3 cells-09-00463-f003:**
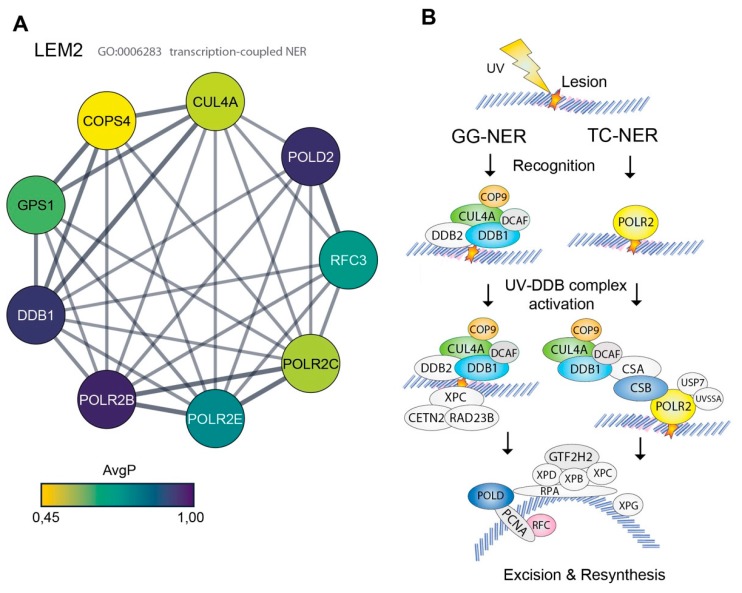
The LEM2 interactome contains several proteins involved in the nucleotide excision repair. (**A**) The identified high confidence interactors of LEM2 of indicated GO processes were further grouped and analyzed using the STRING database. The fill color of the nodes represents the AvgP, and the line width the STRING interaction score. (**B**) Schematic representation of nucleotide excision repair (NER) pathway. GG-NER, global genomic-NER; TC-NER, transcription coupled NER. Proteins biotinylated by BirA*-LEM2 are colored; proteins not found in the LEM2 interactome are light gray.

**Figure 4 cells-09-00463-f004:**
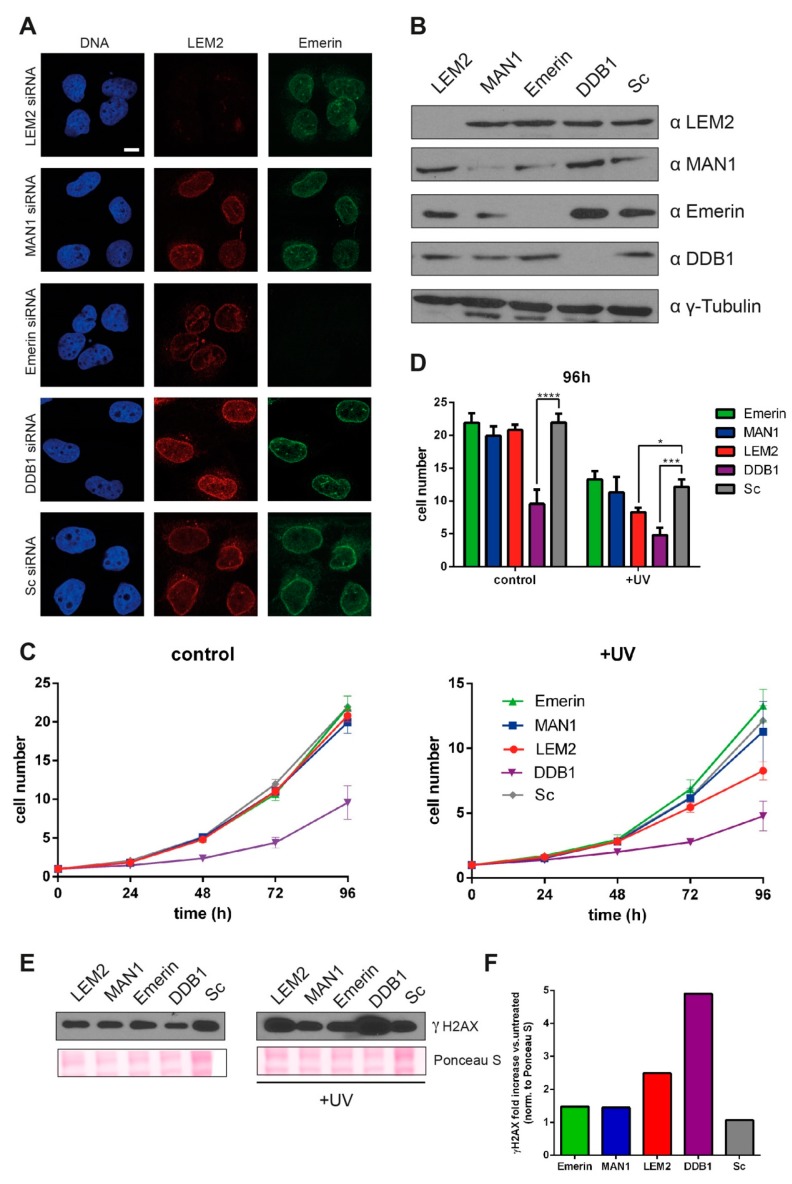
LEM2 knockdown impairs DNA damage response after UV-C irradiation. U2OS cell viability was accessed following knockdown of emerin, MAN1, LEM2 or DDB1 by RNA interference. Scrambled RNA (Sc) was used as negative control. (**A**) Immunofluorescence analyses of U2OS cells 48 h after transfection with indicated siRNAs stained for DNA (blue), LEM2 (red) and emerin (green). Bar, 10 µm. (**B**) Immunoblot analysis of whole-cell lysates 48 hours after transfection with indicated siRNAs using antibodies as indicated on the top. γ-Tubulin served as loading control. (**C**) Emerin- MAN1-, LEM2- and DDB1-depleted cells or cells treated with scrambled RNA were left untreated (control) or irradiated with a sublethal dose of UV-C (5 J/m^2^) (+UV) and cell proliferation was analyzed for 96 h postirradiation. (**D**) Bar graph representing cell numbers at the 96 h time point. Statistical analysis was performed using two-way ANOVA and Tukey’s multiple comparison, n = 8, *, *p*-values: ‘*’ for *p* < 0.05; ‘**’ for *p* < 0.01; ‘***’ for *p* < 0.001; ‘****’ for *p* < 0.0001. Error bars represent standard error of the mean. (**E**) Whole cell lysates were analyzed by immunoblotting before and 48 h after UV treatment, using antibodies to phosphorylated H2AX (γH2AX). Ponceau S stain served as loading control. (**F**) Quantification of immunoblot. Graph represents fold increase of γH2AX after UV treatment normalized to total protein.

**Table 1 cells-09-00463-t001:** High confidence interactors in enriched biological processes. Mass spectrometric data were analyzed using SAINT v. 2.5.0 [43,44] based on an average probability of ≥ 0.45. Gene ontology (GO) enrichment analysis for each LEM protein was done by using ENRICHR considering a fold change ≥ 1 and *p*-value ≤ 0.05 [47,49]. AvgP: Average Probability.

**EMERIN** **GO: 1902749** **G2/M Transition of Mitotic Cell Cycle**			
**Gene ID**	**Gene Symbol**	**Protein Name**	**AvgP**
22994	CEP131	**Centrosomal protein of 131 kDa**	0.708
9662	CEP135	**Centrosomal protein of 135 kDa**	0.4503
9928	KIF14	Kinesin-like protein KIF14	0.452
5347	PLK1	Serine/threonine-protein kinase PLK1	0.4543
**MAN1** GO: 0022618 ribonucleoprotein complex assembly			
**Gene ID**	**Gene Symbol**	**Protein Name**	**AvgP**
988	CDC5L	Cell division cycle 5 like	0.4753
7536	SF1	Splicing factor 1	0.4905
9092	SART1	Spliceosome associated factor 1	0.466
27339	PRPF19	Pre-mRNA processing factor 19	0.623
22938	SNW1	SNW domain containing 1	0.473
**LEM2** GO: 0006283 transcription-coupled NER			
**Gene ID**	**Gene Symbol**	**Protein Name**	**AvgP**
51138	COPS4	COP9 signalosome complex subunit 4	0.4543
8451	CUL4A	**Cullin-4A**	0.5058
1642	DDB1	DNA damage-binding protein 1	0.816
2873	GPS1	**COP9 signalosome complex subunit 1**	0.5923
5425	POLD2	DNA polymerase delta subunit 2	0.8263
5431	POLR2B	DNA-directed RNA polymerase II subunit RPB2	0.8398
5432	POLR2C	DNA-directed RNA polymerase II subunit RPB3	0.5248
5983	RFC3	Replication factor C subunit 3	0.4825

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
