# Peer review of "Comparative Interactome Analysis of Emerin, MAN1 and LEM2 Reveals a Unique Role for LEM2 in Nucleotide Excision Repair"

_cells, 2020, doi:10.3390/cells9020463_

Round 1

Reviewer 1 Report

The original work by Moser et al entitled, “Comparative interactome analysis of emerin, MAN1 and LEM2 reveals a unique role for LEM2 in nucleotide excision repair” describes how proximity based labeling identifies interactomes for each of the inner nuclear membrane proteins emerin, MAN1 and LEM2. Here the authors identify a specific enrichment of the nucleotide excision repair (NER) pathway associated with LEM2, with over representation of multiple molecules in this Gene Ontology Consortium-defined signaling cascade. The work overall presents some novel findings, but a few outstanding items regarding experimental design and data interpretation tempers the originality of the study. 

Major comments/questions:

Was a “BioID-only” construct used as a control to account for random BioID labeling? While the use of an inducible system allows for temporal control of the various BioID fusion proteins, “BioID-only” as well as an N144-BioID fusion protein (Birendra KC et al, Mol Biol Cell 2017) can allow for spatial controls to diminish background. The choice of a transformed cell line, particularly with respect to nucleotide excision repair and DNA metabolism in general may exhibit different cellular physiology than untransformed counterparts. Would the repair pathway enrichment be expected to be as enriched in a regular cell? The authors should include this in their Discussion. The authors note that the stringency of the bioinformatic analysis eliminated the expected nuclear envelope components such as the nuclear lamina and nuclear pore complexes from the final enrichment. Perhaps inclusion of the controls mentioned in point #1 (see above) may allow relaxation of the stringency to avoid possible false negative findings.

Minor comments:

Figure 1C, left panel - please correct spelling of doxycycline Figure 1C, right panel - is there an explanation for the non-nuclear staining observed in doxycycline treated cells? The BioID fusion proteins are all inner nuclear membrane proteins, yet some nucleoplasmic foci and cytoplasmic staining can be observed for all three.

Reviewer 2 Report

Moser et al. studied the interactions of nuclear envelop proteins (Emerin, MAN1 and LEM2) with additional partners at the nuclear envelop. For that purpose, they have utilized the BioID approach that detects proximity of proteins, within ~20 nm. Followed by purification and MS analysis the authors identified a large number of interacting partners for these proteins. Most interestingly, the authors identified that LEM2 interacts with DNA repair proteins (while Emerin and MAN1 not), involved in nucleotide excision repair. Therefore, experiments delineate these findings were conducted and confirmed the physiological relevant of this finding.   The work presented here is elegant and would fit nicely to this journal. Thus, I would recommend to accept the paper for publication in Cells.

Minor poinr:

In this work the authors used inducible expression construct. However, were the expression levels of the Bir-A protein conjugates similar to the physiological levels ? A comparison between the endogenous and the induced protein would be informative.

Reviewer 3 Report

This manuscript by Moser et . al. describes the application of BioID to three transmembrane LEM-domain proteins that enrich in the inner nuclear membrane. BioID experiments were performed in U2OS cells with dox inducible expression of the BioID proteins. Controls were cells not exposed to dox. The candidates were compared and GO analysis was used to identify functional subsets of the candidates. LEM2 was would to have one such functional subset related to the nucleotide excision repair pathway. When LEM2 was knocked down the cells became more sensitive to DNA damage and/or were partially deficient in repair.

Overall, there are significant concerns with the lack of an appropriate control for the BioID experiments. There is also an even more significant concern in how the BioID results were used to draw biological conclusions. The limited extent of data concerning DNA damage/repair is not sufficient to rescue the other concerns.

Specific comments: 

Fig 1B image appears partially cut off

Fig 1D: Could a loading control be included to allow appropriate interpretation of the results. And since the data is presented as three independent blots, or at least one blot cut into three sections, can the authors provide assurance that these were from the same blot and taken at the same exposure.

Fig 1C. The IF images would suggest that there is a nucleoplasmic localization of the LEM-domain fusion proteins as seen by anti-V5 since the nucleolar exclusion is clearly evident. If there were not confocal images this might be understandable, but since the methods indicate confocal microscopy was used why is there a nucleoplasmic localization of the V5 tag/BioID fusion proteins?

A BirA*-alone or other similar control was not utilized. BirA*-alone or similar has become a standard requirement for BioID experiments as it allows exclusion of proteins that have affinity to the biotin ligase, of which there are many. For example, AHNAK is a highly abundant protein detected in these studies and is abundantly detected by the ligase alone for LEM2 and MAN1. It could a legitimate partner or could be a ligase mediated background protein.

Why is HSPA5 so abundantly detected by LEM2? It is an ER lumen protein with no clear localization in the cytoplasm or nucleoplasm. Similar for UGGT1, PRKCSH, GANAB, CALR, FAF2, and likely many others. Seems like a lot of ER lumen proteins were detected despite the ligase being on the other side of the membrane. This raises suspicion of contamination or background.

The relative number of detected candidates is not remotely similar between the compared BirA*-fusion proteins. There are 647, 209 and 96 high confidence candidates detected by LEM2, MAN1 and Emerin, respectively for a ratio of 7:2:1. And the sum of spectral counts is 17426, 9041, 2615 so this doesn’t reflect a lot more of the same proteins detected by Emerin compared to the others. Instead it likely reflects more proteins/peptides being analyzed by MS for some compared to others.  Thus, the numbers of unique proteins for each is not surprisingly 479, 52 and 31 for each of those proteins. It is therefore disingenuous to consider the vast majority of the LEM2 candidates as unique since it simply reflects a better pulldown, MS analysis and/or expression level of the fusion protein that led to all those unique proteins. Perhaps the authors could threshold for the top 90-96 candidates ranked by spectral detection to do comparative analysis so that there are similar numbers of identified proteins, or perhaps use some kind of similar total spectral count thresholding. If you use the top 96 proteins as a cut off you still would get ~64, 44 and 70 unique proteins for LEM2, MAN1 and Emerin, respectively. These could be informatically compared with reasonable confidence.

Caution should be used with GO terms as cellular functions and protein interactions do not always go hand-in-hand. For example, SMAD3 may interact with USP15, but WWOX regulation of TGFb signaling may happen out in the cytoplasm and may not be a physical SMAD interaction, but would function spatially somewhere else in the signaling pathway.

DDB1 was the 173rd most abundant unique protein detected on the LEM2 candidate list. The other NER components are even less abundant. It is difficult to see how this leads to a conclusion of considerable interaction, especially when these proteins could well be detected by MAN1 or Emerin if a similar level of proteins had been detected by those BirA* fusion proteins. This doesn’t nullify any results from figure 4 but undercuts the rationale for doing those experiments.

That there is more overlap between LEM2 and MAN1 than with Emerin would be expected given the numbers of identified candidates and doesn’t reveal anything useful.

Round 2

Reviewer 1 Report

The revised manuscript satisfactorily addresses all concerns raised previously.

Minor:

On pg. 5, line 189 – please correct spelling of ‘Doxicycline’ On pg. 8, line 277 – please correct spelling of ‘comforms’

Reviewer 3 Report

The authors have addressed my major concerns, most notably by incorporating an appropriate control for their BioID experiments. I remain surprised by the relative paucity of known nuclear envelope proteins in their results, including well characterized binding partners and have seen, admittedly unpublished, BioID results from some of these same proteins with a more conventional abundance of lamins and other INM proteins as the top candidates. Emerin is perhaps the most striking as there is no LMNA, a well know binding partner, or TMPO, LEMD3 or Sun2 which I have seen as  top candidates on other BioID-emerin experiments. That said, the results are what they are and the approaches seem appropriate. Perhaps the authors could address the relative lack of expected nuclear envelope-associated proteins where appropriate in the manuscript.